# Beyond Abstracts:
# Learning Scientific Paper Embeddings from Full-Text Windows

Younes Djemmal[1]    Oloruntobi Olutola[1, 2]    Kim Gerdes[1, 2]
(1) Questel, Paris, France
(2) Université Paris-Saclay, LISN, CNRS, France
ydjemmal@questel.com, oloruntobi.olutola@lisn.fr, kim.gerdes@lisn.fr

## RÉSUMÉ

Les plongements d'articles scientifiques sont généralement entraînés uniquement sur le titre et le résumé. Nous présentons S2Full, un corpus d'entraînement de 2,52 millions d'articles en texte intégral avec liens de citation, et entraînons sur des fenêtres aléatoires du corps avec un objectif d'auto-alignement. Sur un benchmark de recherche d'articles fondé sur les citations, notre approche présente de bonnes performances lors de la recherche d'articles en utilisant uniquement les titres et résumés, tout en améliorant les résultats lorsque les requêtes incluent du contenu issu du corps principal des articles. Les meilleurs scores observés proviennent d'un modèle ModernBERT, tandis que la comparaison SciBERT pleinement entraînée montre déjà le même effet directionnel. L'entraînement sur le texte intégral améliore ainsi la recherche citationnelle à partir de formulations plus variées du contenu d'un article, au-delà de son résumé.

## ABSTRACT

**Beyond Abstracts: Learning Scientific Paper Embeddings from Full-Text Windows**

Paper embeddings are typically trained on titles and abstracts only. We present S2Full, a 2.52M-paper full-text training corpus with citation links, and train on random body windows with a self-alignment objective. In a citation-based article retrieval benchmark, our approach maintains strong performance when retrieving papers using only titles and abstracts, while improving retrieval when queries include content from the main body of papers. The strongest observed scores come from a ModernBERT model, while the fully trained SciBERT comparison already shows the same directional gain from body-view exposure. Full-text training, therefore, improves citation-based retrieval from more diverse formulations of a paper's content beyond its abstract.

MOTS-CLÉS : plongements d'articles, texte intégral, fenêtres aléatoires, recherche citationnelle, SBERT.

KEYWORDS: paper embeddings, full text, random windows, citation-based retrieval, SBERT.

# 1  Introduction

Scientific paper embeddings are central to citation-based retrieval, recommendation, and citation analysis, but widely used citation-informed models such as SPECTER, SciNCL, and SPECTER2 remain centered on title-abstract representations rather than full-text body sections (Cohan *et al.*, 2020; Ostendorff *et al.*, 2022; Singh *et al.*, 2023). The practical reason is architectural: widely used sentence-transformer encoders based on BERT are limited to short contexts, so the introduction,

methods, results, and discussion of a paper are usually ignored during training.

We address this mismatch by training on *random full-text windows* rather than on title and abstract alone. Instead of forcing an entire article into a single short input, we sample windows from different body sections and train the encoder to align citation-linked papers while also aligning multiple views of the same paper. This preserves the standard sentence-transformer setup while exposing the model to much more of the document. Our evaluation target throughout is citation-based paper retrieval rather than full user search relevance: citation links operationalize one useful notion of scholarly proximity here, but they remain imperfect proxies for semantic relatedness and user relevance.

Our contributions are as follows:

1. We build **S2Full**[1], a large-scale corpus combining structured full text from peS2o (Allen Institute for AI, 2023) with citation data from the Semantic Scholar Open Data Platform (Kinney *et al.*, 2023): a final training set of 2.52M papers, 4.76M citation pairs, and 34M body sections across 19 scientific domains, together with a section extraction pipeline and a held-out benchmark of 10,022 queries.

2. We propose a **random-window training** method that samples fixed-length windows from body sections and combines citation-contrastive learning (80% of each batch) with a self-alignment objective that aligns different views of the same paper (20%). We design and compare four sampling strategies encoding different retrieval assumptions, and show that the simplest (**both_random**) is optimal.

3. Within a fixed contrastive recipe, we show that exposing the encoder to body windows **preserves abstract-level retrieval** while **improving citation-based body retrieval**. The effect is consistent across both encoder families (SciBERT and ModernBERT).

4. We demonstrate that our models produce a **well-spread embedding space** that supports both abstract-level and section-level retrieval, whereas SPECTER2's embeddings occupy a much narrower region of the representation space. This geometric quality, measured through alignment and uniformity metrics, helps explain the retrieval gap: discriminating relevant from irrelevant documents requires pushing negatives apart, which a concentrated space cannot do effectively.

5. We find that **training-matched windowed body representations retrieve better than raw sections** on the citation-based benchmark, indicating that short or uneven sections underuse the encoder context and benefit from surrounding text.

These results suggest that full-text-trained embeddings improve citation-based retrieval from a broader range of article formulations, though this still needs to be validated beyond citation-derived relevance.

## 2   Related Work

Citation-informed scientific embeddings were established by SPECTER (Cohan *et al.*, 2020) and extended by SPECTER2 (Singh *et al.*, 2023) and SciNCL (Ostendorff *et al.*, 2022). These methods

---

[1]The S2Full dataset is publicly available for research purposes at https://huggingface.co/datasets/younesdjemmal/S2Full.

are effective, but they remain centered on title and abstract. At the same time, sentence-transformer architectures (Reimers & Gurevych, 2019) impose short context limits that make direct full-article encoding difficult. This motivates training schemes that expose the encoder to multiple local views rather than a single truncated input. We therefore combine citation-based contrastive learning with random body-window sampling, and compare a standard scientific encoder (SciBERT (Beltagy *et al.*, 2019)) with a longer-context encoder (ModernBERT (Warner *et al.*, 2025)). The present comparison targets citation-based paper retrieval under this embedding setup rather than full retrieval pipelines built from lexical, hybrid, or reranking systems.

# 3 Dataset and Method

## 3.1 S2Full Corpus

S2Full combines paragraph-structured full text from peS2o with paper metadata and citation links from Semantic Scholar. The goal is to build a training resource in which title, abstract, and body text can all be sampled while preserving a large citation graph for contrastive supervision.

We first retain only papers with a venue and a field of study, then keep citation edges whose endpoints both survive this filter. We remove self-citations, deduplicate citation pairs, filter out very high-degree hub papers, and require enough usable body text to sample meaningful section windows. Finally, we cap the largest domains so that the corpus is not overwhelmingly dominated by biomedical literature. This domain cap is an engineering choice meant to improve cross-domain comparability rather than to match the true distribution of scientific production. More broadly, these filters are not cosmetic: they define the retrieval task and reduce noise in the citation graph by removing documents that would otherwise contribute little usable text or create problematic high-degree neighborhoods for contrastive sampling (Ostendorff *et al.*, 2022; Xu *et al.*, 2025). They also make the resulting corpus less representative of the full scholarly record.

After filtering, the final training corpus contains 2.52M papers and 4.76M citation pairs across 19 scientific domains, yielding 34M body sections. The held-out benchmark contains 10,022 queries, 63,095 unique papers, and 55,030 citation pairs with zero overlap with the training papers. Table 1 shows how the corpus is constructed; section extraction details are given in Appendix A.1 and the preprocessing filters are discussed in Appendix A.2.

| Filtering stage | Papers | Pairs |
|---|---:|---:|
| Semantic Scholar metadata | 6,282,865 | — |
| + venue filter | 5,193,622 | — |
| + fields-of-study filter | 4,545,158 | — |
| Raw citation edges (both endpoints valid) | — | 27,310,848 |
| + bidirectional dedup + self-citation removal | — | 22,171,946 |
| + high-degree filter ($> 200$ pairs) | — | 20,628,139 |
| + text quality filter | 3,602,952 | 18,011,917 |
| + domain cap $\rightarrow$ training set | 2,516,974 | 4,762,877 |
| Test split | 63,095 | 55,030 |

Table 1: S2Full dataset construction funnel: paper and pair counts at each filtering stage.

## 3.2 Training Views and Objectives

We define two types of text units per paper: a single **TA** view (title+abstract, concatenated) and multiple **body views**, each corresponding to a detected section span. During training, a body view is converted into a fixed-length window beginning at a section header and extended forward to a target length. We use 358 words for SciBERT and 716 words for ModernBERT, corresponding to roughly 70% of the encoder context budget.

On top of the base encoder, we use a sentence-transformer style setup with mean pooling and a two-layer projection head. We train both SciBERT and ModernBERT variants so that the comparison tests not only the sampling strategy but also the effect of longer context, since ModernBERT is explicitly designed for efficient long-context encoding and has been validated on long-document understanding and retrieval settings (Warner *et al.*, 2025; Jamalpur & Maram, 2025; Lee *et al.*, 2025).

## 3.3 Pair Sampling Strategies

Each training example consists of an anchor view and a positive view drawn from a citation pair $(p_a, p_+)$. We experiment with four strategies that differ in which text unit type is assigned to each role, encoding a different retrieval assumption.

- **both_random**: anchor and positive are each drawn uniformly from all eligible views of their respective papers. This is the most general strategy, training the model to align any view type from one paper with any view type from a related paper.

- **ta_ta**: both anchor and positive are always TA views. This provides an abstract-level baseline that isolates the contribution of the large-batch in-batch negative framework on abstract-only data, without using any full-text signal.

- **anchor_random_pos_ta**: the anchor is drawn randomly while the positive is always a TA view. This asymmetric strategy trains the model to retrieve papers by their abstract when the query is a full-text section.

- **no_ta_ta**: anchor is drawn randomly; if the anchor is a TA view, the positive is forced to be a body view, preventing TA–TA co-occurrence. This maximises cross-granularity alignment without the asymmetry of **anchor_random_pos_ta**.

SciBERT is trained under all four strategies. ModernBERT is trained under **both_random** and **ta_ta** only, covering the two most contrasting strategies for a long-context baseline comparison.

**Self-alignment augmentation.**    In addition to citation pairs, 20% of each batch consists of *same-paper pairs*: two distinct views drawn from the same paper and used as a positive pair. This operationalises the self-alignment objective: the model is rewarded for assigning high similarity to, for example, an introduction and a results section from the same paper, independently of citation structure. The augmentation is applied uniformly across all four strategies.

## 3.4  Training Objective

We train with CachedMultipleNegativesRankingLoss (Henderson *et al.*, 2017; Gao *et al.*, 2021a), using large in-batch negatives to discriminate each positive pair from the rest of the batch. This follows the standard contrastive setup in which the other examples in the mini-batch act as negatives (Gao *et al.*, 2021b). In practice, each paper view is therefore learned not only against its paired positive example, but against a large set of other scientific papers seen in the same update. The present experiments isolate the effect of body-view exposure within this fixed recipe, but they do not separately ablate the self-alignment ratio, projection head, or context length. The full hyperparameter table remains in Appendix B.

# 4  Experimental Setup

## 4.1  Test Set

We evaluate on the held-out S2Full test split: 10,022 anchor papers, 63,095 total papers, and 55,030 citation pairs with zero paper overlap with the training set. The held-out benchmark is deliberately stricter than the validation setup used during training: no paper that appears in the test set, whether as a query or as a relevant neighbor, appears in the training data. The task therefore measures generalization to unseen papers rather than memorization of citation links, although its relevance labels remain proxy labels derived from citation edges rather than direct query-need judgments.

For cross-model comparability, one body section per paper is pre-selected deterministically (seeded by paper ID) and frozen in a shared *paper log* reused by all models, so that every model is evaluated on exactly the same body text for each paper. This matters because section choice changes difficulty: some sections are concise and topic-heavy, while others contain formulas, experimental detail, or boilerplate.

## 4.2  Baseline

We compare against SPECTER2 (Singh *et al.*, 2023) with the proximity (PRX) adapter, the current state of the art for citation-based scientific document embeddings. SPECTER2 shares the same SciBERT encoder architecture and 512-token context limit as our SciBERT models, making the comparison architecture-controlled. We do not further apply the same training pipeline to SPECTER2, as it is already trained under a comparable title-and-abstract retrieval setting using the same SciBERT backbone and a similarly scaled training corpus, making it an appropriate baseline for comparison.

## 4.3  Body Representation Modes

We evaluate body sections under two representation modes:

**Full section.**   The complete text between consecutive section headers, used as-is. Section lengths vary widely: the median is 339 words and 51.8% of sections are shorter than 358 words (the SciBERT

training target), meaning more than half of queries underutilise the encoder's context window.

**Windowed.** A fixed-length passage starting at the section header, filled to $0.7 \times L_{\max}$ words (358 for SciBERT, 716 for ModernBERT) by continuing past the section boundary into subsequent text, matching the training-time extraction (Section 3.2). 91.9% of windowed extractions reach the full 358-word target for SciBERT; the remaining 8.1% correspond to sections near the end of the paper where insufficient text remains. Table 2 summarises the length distributions.

| Mode | Mean | Median | Q25 | Q75 | Max |
|------|------|--------|-----|-----|-----|
| Full section | 606 | 339 | 157 | 770 | 22,009 |
| Windowed (SciBERT) | 344 | 358 | 358 | 358 | 358 |
| Windowed (ModernBERT) | 661 | 716 | 716 | 716 | 716 |

Table 2: Word-count statistics for body section representations across the 63,095 test papers. Windowed extraction fills short sections by continuing past section boundaries, concentrating lengths tightly at the target.

**Note on ModernBERT full-section evaluation.** Although ModernBERT was trained with a maximum sequence length of 1,024 tokens, the full-section evaluation uses the model's native 8,192-token context window. This means ModernBERT can encode full sections without truncation in most cases, unlike SciBERT which truncates at 512 tokens.

The main comparison uses *windowed* body representations because they match the training distribution. The full-section ablation is reported in Section 5.2.

## 4.4   Retrieval Scenarios

We evaluate three core retrieval scenarios:

- **TA→TA**: both query and corpus are title-abstract views. Standard abstract-level retrieval, measuring compatibility with prior work.

- **Body→TA**: the query is a body section; the corpus consists of title-abstract views. This cross-granularity scenario tests whether a section-level query can retrieve the correct paper by its abstract.

- **Body→Body**: both query and corpus are body sections, measuring section-level retrieval.

Unlike the training validation protocol, the test evaluation excludes the query paper from its own retrieval results, so that a body query cannot retrieve its own paper's TA. All reported metrics therefore reflect strictly cross-paper retrieval performance.

## 4.5 Metrics

The primary metric is NDCG@10, with MRR and Recall@$k$ ($k = 1, 10, 100$) reported in the appendix. In addition to retrieval metrics, we report three representation-quality metrics that characterise the embedding space geometry: pair alignment, uniformity (Wang & Isola, 2020), and intra-article alignment; their definitions are given in Appendix 5.3.

## 4.6 Training Setup

All models are trained using the same contrastive learning framework to ensure that observed differences arise primarily from the encoder architecture and the view-sampling strategy. Both base encoders, SciBERT and ModernBERT, converged rapidly and reached stable validation performance early in training. All SciBERT variants were allowed to complete the full three training epochs. For ModernBERT, only two representative sampling strategies were evaluated, and training was stopped earlier because its substantially larger context window increased computational cost. Given the early convergence observed for both encoders, the additional gains expected from completing all three epochs for the ModernBERT variants were likely to be marginal.

| Hyperparameter | SciBERT | ModernBERT |
|---|---|---|
| Max sequence length | 512 | 1,024 |
| Target body window | 358 words | 716 words |
| Embedding dimension | 4,096 | 4,096 |
| Batch size $N$ | 2,048 | 2,048 |
| Mini-batch size $M$ | 128 | 64 |
| Learning rate | $5 \times 10^{-5}$ | $5 \times 10^{-5}$ |
| Epochs | 3 | $\approx 1.8$ |
| Precision | bfloat16 | bfloat16 |
| Strategies trained | 4 | 2 |

Table 3: Main training settings for the two encoder families. The full hyperparameter listing remains in the appendix.

# 5 Results and Discussion

Throughout this section, **SB** denotes SciBERT and **MB** denotes ModernBERT. Bold values indicate the best result per column. Additional evaluation details (metric definitions, prefix handling) are given in Appendix C.

## 5.1 Retrieval Results

Table 4 is the core result. On TA→TA, all our models outperform SPECTER2 by +10 to +12.5 NDCG@10 points (full metrics in Appendix G). **ta_ta** training is best (0.544), as expected because training and evaluation distributions match exactly, but **both_random** is nearly identical (0.535):

exposing the model to body text does not degrade abstract-level retrieval. On body-based citation retrieval, the picture reverses sharply. Body→Body is the hardest task because it requires recognizing a paper's contribution from the partial signal of a single section window.

The clearest causal comparison across all four strategies comes from SciBERT, where the three body-aware strategies cluster around 0.439–0.441 on Body→TA, whereas **ta_ta** falls to 0.406. On Body→Body the gap is wider still: 0.366–0.391 vs. 0.304. Notably, even SB **ta_ta** (0.406) outperforms SPECTER2 (0.273) by +13.3 points on Body→TA despite being trained only on abstracts, showing that a substantial part of the improvement comes from our training framework itself (large-batch in-batch negatives, self-alignment augmentation). Body-text training provides a further +3.3 points on top of these framework gains. Even our weakest variant (SB ta_ta on Body→Body, 0.304) exceeds SPECTER2 (0.192) by +11.2 points. MB **ta_ta** (0.454) surpasses all SciBERT body-trained models on Body→TA, suggesting that ModernBERT's encoder capacity provides a stronger foundation for cross-granularity transfer.

Among body-aware strategies, **both_random** and **no_ta_ta** are nearly identical, showing that explicitly blocking TA–TA citation pairs adds no benefit when the natural probability of drawing two TA views is already low (∼2% of citation pairs, Table 8). **anchor_random_pos_ta** is strong on Body→TA (0.441) but weaker on Body→Body (0.366) because constraining the positive to always be a TA view means body; body alignment is learned only through the 20% self-alignment augmentation. All strategies, including **ta_ta**, receive self-alignment pairs with body views, which explains why **ta_ta** still achieves non-trivial body retrieval (0.304 Body→Body) despite never seeing body text in its citation pairs. **both_random** offers the best overall tradeoff between simplicity and performance; full per-metric Body→TA and Body→Body results are in Appendix E.

| Model | TA→TA | Body→TA | Body→Body |
|---|---|---|---|
| SB both_random | 0.518 | 0.439 | 0.391 |
| SB anchor_r._pos_ta | 0.528 | 0.441 | 0.366 |
| SB no_ta_ta | 0.517 | 0.439 | 0.389 |
| SB ta_ta | 0.537 | 0.406 | 0.304 |
| MB both_random | 0.535 | **0.485** | **0.457** |
| MB ta_ta | **0.544** | 0.454 | 0.358 |
| SPECTER2 | 0.419 | 0.273 | 0.192 |

Table 4: NDCG@10 across retrieval scenarios using windowed body representations. Bold indicates best overall per column. TA→TA favors **ta_ta**, while the body-based citation retrieval settings favor body-aware training.

## 5.2  Why Windowed Extraction Matters

The main results use windowed body representations because training also uses fixed-length body windows. This design choice is important rather than incidental. If we instead feed raw full sections at test time, retrieval quality drops consistently for all models, including ModernBERT.

Table 5 shows the pattern. For SciBERT, windowed extraction improves Body→TA by about 4 points and Body→Body by about 6 to 7 points. For ModernBERT, the gain is even larger: **both_random** rises from 0.413 to 0.485 on Body→TA and from 0.337 to 0.457 on Body→Body. The likely reason

is simple: many raw sections are shorter than the model's target input length, whereas windowing fills the encoder with a more informative and more training-consistent chunk of text. Note that the ModernBERT full-section evaluation uses the full 8,192-token context, not the 1,024-token training limit; even with this additional capacity, windowed extraction still outperforms, likely because many raw sections remain shorter than the target window length regardless of the encoder's capacity.

| Model | Body→TA | | | Body→Body | | |
|---|---|---|---|---|---|---|
| | Full | Wind. | $\Delta$ | Full | Wind. | $\Delta$ |
| SB both_random | 0.396 | 0.439 | +4.3 | 0.322 | 0.391 | +6.9 |
| SB anchor_r._pos_ta | 0.397 | 0.441 | +4.4 | 0.297 | 0.366 | +6.9 |
| SB no_ta_ta | 0.396 | 0.439 | +4.3 | 0.322 | 0.389 | +6.7 |
| SB ta_ta | 0.363 | 0.406 | +4.3 | 0.241 | 0.304 | +6.3 |
| MB both_random | 0.413 | **0.485** | +7.2 | 0.337 | **0.457** | +12.0 |
| MB ta_ta | 0.383 | 0.454 | +7.1 | 0.252 | 0.358 | +10.6 |
| SPECTER2 | 0.247 | 0.273 | +2.6 | 0.161 | 0.192 | +3.1 |

Table 5: NDCG@10 for full-section vs. windowed body representations. $\Delta$ = absolute point difference ($\times 100$). ModernBERT full-section evaluation uses the native 8,192-token context, not the 1,024-token training limit. Windowing improves all models, with the largest gains on Body→Body.

The aggregate scores are not driven by a single field: our models outperform SPECTER2 across all 19 scientific domains, with gains ranging from +14.1 points (History) to +31.8 points (Art), and the ranking of domains is broadly stable across model families (see Appendix D).

## 5.3 Representation Quality

Beyond retrieval scores, we examine the geometry of the embedding space using alignment, uniformity (Wang & Isola, 2020), and intra-article alignment (definitions in Appendix F). Table 6 reports these metrics on windowed embeddings.

| Model | Align ($\downarrow$) | Unif ($\downarrow$) | IntraAl ($\downarrow$) |
|---|---|---|---|
| SB both_random | 0.321 | $-0.918$ | 0.164 |
| SB anchor_r._pos_ta | **0.259** | $-0.645$ | 0.164 |
| SB no_ta_ta | 0.323 | $-0.925$ | **0.163** |
| SB ta_ta | 0.295 | $-0.691$ | 0.211 |
| MB both_random | 0.355 | $-\textbf{0.996}$ | 0.162 |
| MB ta_ta | 0.367 | $-0.981$ | 0.255 |
| SPECTER2 | $0.088^\dagger$ | $-0.078^\dagger$ | $0.063^\dagger$ |

Table 6: Representation quality metrics (windowed mode). All: lower is better. Bold = best among models with well-spread embedding spaces. † SPECTER2's alignment and intra-article alignment are numerically lowest but reflect a concentrated embedding space (uniformity $\approx 0$), not discriminative structure (see text).

SPECTER2 obtains the numerically lowest alignment and intra-article alignment, but these values are

misleading: its uniformity of $-0.078$ reveals that nearly all embeddings occupy a small region of the hypersphere. In such a concentrated space, every pair of embeddings is close, so alignment and intra-article alignment are trivially minimized without the model having learned any discriminative structure. This is consistent with its weak retrieval scores: a retrieval system needs to push irrelevant documents *away* from the query, which requires a well spread-out embedding space. Our models, by contrast, achieve uniformity values between $-0.645$ and $-0.996$, indicating that the embedding space is actively utilized. The **ta_ta** variants show the worst intra-article alignment because their citation pairs never expose the model to body text; **anchor_random_pos_ta** and **ta_ta** achieve the best pair alignment.

# 6 Conclusion

Training on random full-text windows improves scientific paper embeddings on citation-based retrieval benchmarks. The strongest body-retrieval scores come from MB both_random, and the effect is confirmed across both encoder families: body-view exposure improves body-based retrieval without sacrificing TA→TA performance. Analysis of the embedding space further indicates that these gains coincide with a better-spread representation than that of SPECTER2. These results suggest that full-text-trained embeddings can support citation-based retrieval from more diverse formulations of a paper's ideas.

The benchmark and supervision signal are both citation-based, so our results measure retrieval against citation-structure relevance rather than full user relevance; citation links are useful but imperfect proxies for semantic relatedness and may carry disciplinary biases. Several directions remain to be explored. Because our random-window sampling generates a different training set at each epoch, training for more epochs could yield further gains without additional data. Ablating the self-alignment ratio and the role of longer context would help disentangle the contributions bundled in the current recipe, and determining which section types (introduction, methods, results, discussion) carry the strongest signal would refine the sampling strategy. A critical validation step is to evaluate retrieval against actual user search queries, to confirm that the body-level gains translate into practical search improvements; a fuller study should also compare against lexical, hybrid, or reranking baselines.

# Acknowledgements

This work was granted access to the HPC resources of IDRIS under the allocation 2025-AD011016573 made by GENCI.

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

# Appendix

This appendix provides detailed corpus construction, training setup, evaluation protocol, and extended analyses.

# A  Dataset Construction and Statistics

S2Full is constructed by integrating peS2o full text and Semantic Scholar metadata through a shared identifier. We apply venue and field-of-study filters, deduplicate citation edges, remove self-citations, filter high-degree hubs, and require sufficient usable body text. These steps preserve large-scale coverage while avoiding training dominated by survey-like hub papers or documents without meaningful full text, but they also bias the corpus away from parts of the scholarly record that are sparse in metadata, unusually highly cited, or poorly structured in full text. The final training set contains 4.76M citation pairs drawn from 19 domains, while the test set contains 10,022 query papers with zero overlap with training papers.

## A.1  Text Section Extraction

Converting a raw paragraph sequence into structured text sections requires detecting two types of boundaries: section headers and back matter.

**Back-matter detection.** Starting from the end of each document, we mark as back matter any paragraph whose leading text matches known terminal-section signals: *References*, *Acknowledgment(s)*, *Appendix*, *Supplementary material*, *Author contributions*, *Funding*, *Data availability*, *Ethics statement*, and related variants. All paragraphs from the first such marker to the end of the document are excluded from body sections.

**Section header detection.** We use a two-stage heuristic on the remaining body paragraphs. A paragraph is first rejected by a *trash filter* (paragraph too long, ends with sentence-final punctuation, contains table artefacts, etc.). Surviving paragraphs are accepted as section headers if they meet all of the following: at most 150 characters, start with an uppercase letter or digit, no trailing sentence punctuation, and at least one of: fully uppercase text, Title Case, a numbered prefix (1., 2.1, ...), or at most six words. Pre-computed section boundaries are stored alongside each paper in a `chunk_meta` JSON structure.

**Extraction statistics.** Across the 3,602,952 papers that pass the text quality filter, the corpus yields **34,295,771 body sections**. On average, each paper provides 9.5 body sections (median 8), spanning a mean of 3,915 body words (median 3,449). The TA section averages 1,644 characters (median 1,595).

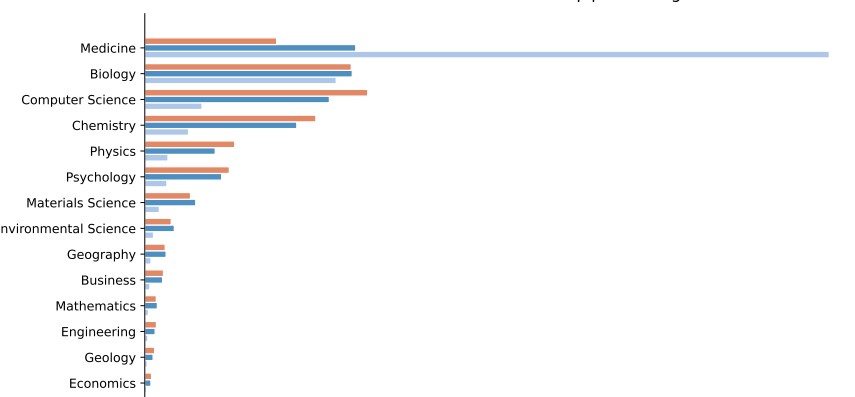

Figure 1: Domain distribution of S2Full at three pipeline stages. The domain cap reduces the dominance of Medicine and Biology and yields a more balanced generalist corpus.

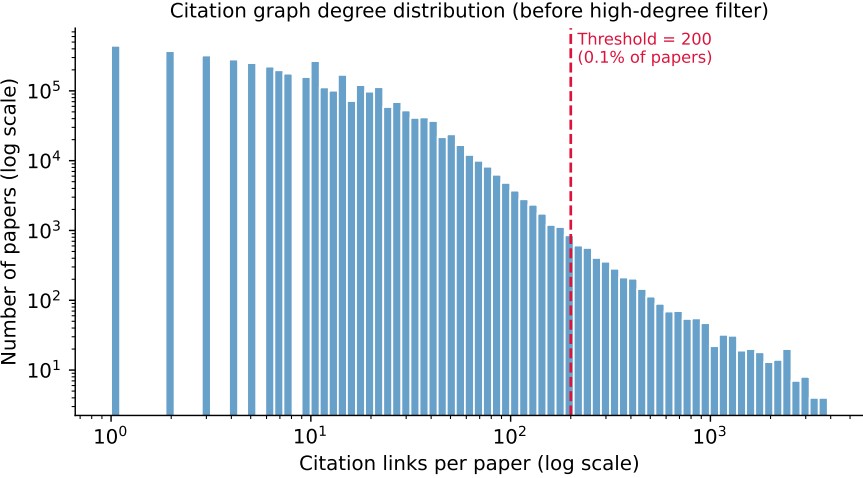

Figure 2: Citation-graph degree distribution before the high-degree filter. The dashed line marks the 200-pair threshold used to remove hub papers.

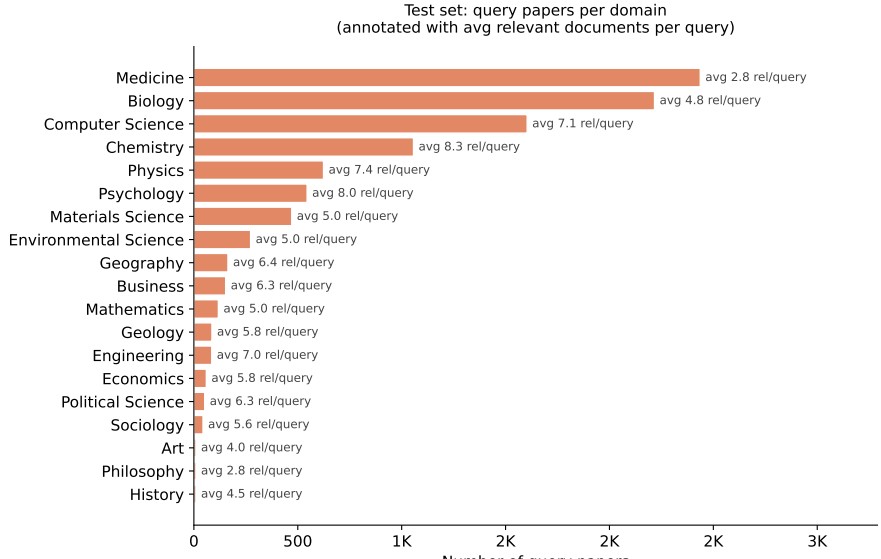

Figure 3: S2Full test split by domain, with the average number of relevant documents per query.

## A.2 Preprocessing and Filtering

We apply a sequence of quality and design filters to the raw data. Table 1 reports the number of papers and citation pairs at each stage.

**Metadata filters.** We restrict to papers that have a non-empty venue field and a non-empty primary field of study. The venue filter removes preprints and low-quality entries that lack a formal publication venue; the field-of-study filter is required for domain assignment and stratification (Section 4.1).

**Citation graph construction.** We load all directed citation edges from the Open Data Platform, retain only edges where both endpoints pass the metadata filters, and remove self-citations. We then apply *bidirectional deduplication*: each pair is normalized by sorting the two paper identifiers lexicographically, and duplicates are removed. This yields an undirected set of citation pairs in which the original citation direction is discarded: the anchor role is determined by lexicographic ordering of paper identifiers, not by which paper cited the other.

**High-degree filter.** We compute each paper's *citation-graph degree* as the total number of dedu-plicated pairs in which it participates (as either endpoint). Papers with a degree exceeding 200 are removed, together with all pairs in which they appear. This threshold targets survey papers and meta-analyses that reference hundreds of works: acting as high-frequency hubs, they would otherwise skew the training distribution and risk causing embedding collapse. As shown in Figure 2, the degree distribution is heavily long-tailed; fewer than 1% of papers exceed the threshold, yet their removal

substantially reduces hub-induced noise.

**Text quality filter.** We retain only papers whose full-text body contains at least 3,000 characters and at least two extractable body sections (Section A.1). This ensures that each paper provides enough content beyond the abstract to allow meaningful body windows to be sampled during training.

**Domain cap.** To prevent the training distribution from being dominated by a small number of fields, we cap each domain at 1,000,000 citation pairs. Without this cap, Medicine (63.9% of pre-cap pairs) and Biology (17.9%) together account for over 80% of training signal. Two domains hit the cap; the remaining 17 are uncapped. The cap preserves the relative order of smaller domains while flattening the top of the distribution, yielding a more balanced generalist corpus (Figure 1).

# B   Training Details

We train sentence-transformer models built on SciBERT and ModernBERT. Each model adds a two-layer projection head and is optimized with CachedMultipleNegativesRankingLoss using large in-batch negatives. Same-paper self-alignment pairs account for 20% of each batch. SciBERT models are trained for 3 full epochs; ModernBERT models are trained for approximately 1.8 epochs. We do not separately ablate the self-alignment ratio, projection head, or context length, so those factors remain bundled with the main comparison.

| Model | Strategy | Steps | Epochs |
|---|---|---|---|
| SciBERT | both_random | 6,837 | 3.0 |
| SciBERT | anchor_random_pos_ta | 6,837 | 3.0 |
| SciBERT | no_ta_ta | 6,837 | 3.0 |
| SciBERT | ta_ta | 6,837 | 3.0 |
| ModernBERT | both_random | 4,190 | ≈1.84 |
| ModernBERT | ta_ta | 4,145 | ≈1.82 |

Table 7: Training runs.

# C   Evaluation Details

**Prefix handling.** All retrieval evaluations are run *without* the [TA]/[BODY] special-token prefixes used during training, to ensure fair comparison with SPECTER2, which was not trained with such prefixes.

**Metric definitions.** All metrics are computed per-query and macro-averaged across queries.

- **NDCG@10** (primary metric): Normalised Discounted Cumulative Gain at rank 10, measuring ranking quality with multiple relevant documents.

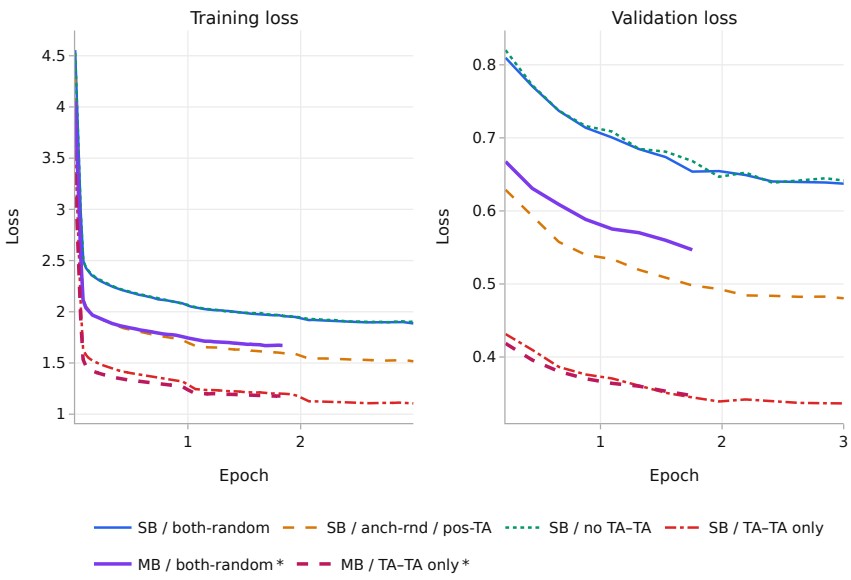

Figure 4: Smoothed training loss curves for all six experimental conditions.

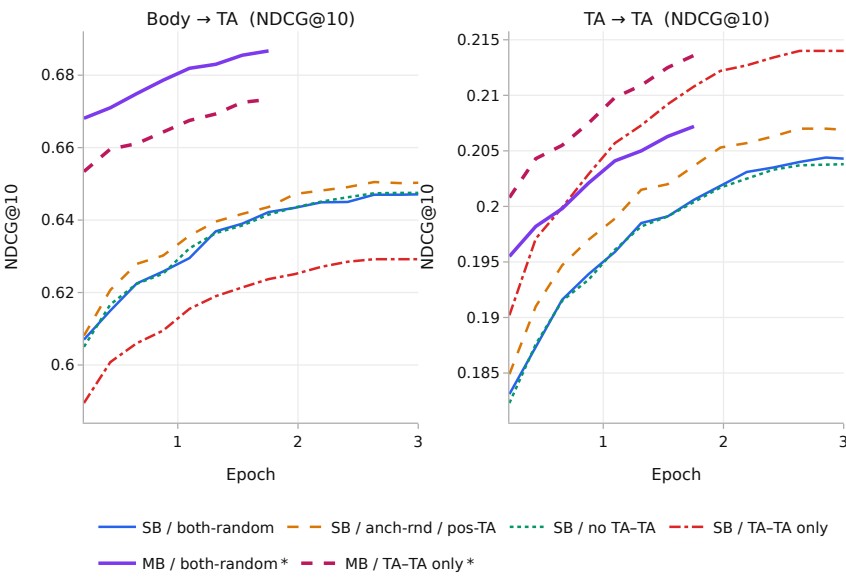

Figure 5: Validation NDCG@10 on Body→TA and TA→TA retrieval during training.

| Strategy | ta–ta | ta–body | body–ta | body–body |
|---|---|---|---|---|
| *Citation pairs (regular)* | | | | |
| both_random | 2% | 11% | 11% | 77% |
| anchor_random_pos_ta | 13% | 0% | 87% | 0% |
| no_ta_ta | 0% | 13% | 11% | 77% |
| ta_ta | 100% | 0% | 0% | 0% |
| *Same-paper pairs (self-alignment, all strategies)* | | | | |
| (all) | 0% | 13% | 13% | 75% |

Table 8: Empirical view-type distribution in SciBERT training batches. Entries give the percentage of pair types observed within a batch for each strategy, split between regular citation-linked positives and same-paper self-alignment pairs. Here, *ta* denotes a title-abstract view and *body* a sampled body-section window, so *ta–body*, for example, means a title-abstract query paired with a body-view target.

- **MRR**: Mean Reciprocal Rank of the first relevant document.

- **Recall**@$k$ ($k = 1, 10, 100$): fraction of relevant documents appearing in the top $k$ results.

**Self-retrieval exclusion.** Unlike the training validation protocol, the test evaluation excludes the query paper from its own retrieval results, so that a body query cannot retrieve its own paper's TA. All reported metrics therefore reflect strictly cross-paper retrieval performance. Training validation included self-retrieval, which inflated absolute Body→TA scores; test-set results are more conservative but more interpretable.

# D  Per-Domain Robustness

Figure 6 shows Body→TA performance broken down by scientific domain. Our models outperform SPECTER2 across all 19 domains, and the ranking of domains is broadly stable across model families: Geology is comparatively easy, while Medicine and Biology remain more difficult. This consistency suggests that the gains from full-text training are not confined to one citation culture or one style of scientific writing. Among large domains ($n > 500$), gains of MB both_random over SPECTER2 range from +18.4 points (Computer Science) to +26.9 points (Physics). Geology achieves the highest absolute NDCG@10 for all models (0.669 for MB both_random), while Medicine (0.430) and Biology (0.453) remain the most challenging, consistent with their broader topical diversity and citation patterns.

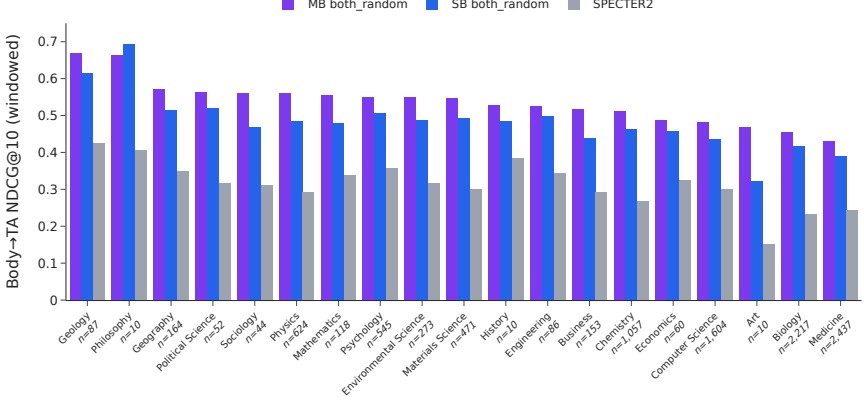

Figure 6: Body→TA NDCG@10 (windowed) per scientific domain, sorted by descending MB both_random score. $n$ = number of test queries per domain.

# E  Extended Results

## E.1  Cross-Granularity Retrieval (Body→TA)

Table 9 reports the full Body→TA retrieval metrics. The three body-aware SciBERT strategies cluster tightly (NDCG@10 0.439–0.441), while **ta_ta** drops to 0.406, confirming that body-text exposure during training is necessary for cross-granularity retrieval. MB both_random leads on every metric with 0.485 NDCG@10.

| Model | NDCG@10 | MRR | R@1 | R@10 | R@100 |
|---|---|---|---|---|---|
| SB both_random | 0.439 | 0.604 | 0.138 | 0.443 | 0.727 |
| SB anchor_r._pos_ta | 0.441 | 0.605 | 0.138 | 0.446 | 0.729 |
| SB no_ta_ta | 0.439 | 0.602 | 0.137 | 0.444 | 0.728 |
| SB ta_ta | 0.406 | 0.569 | 0.127 | 0.412 | 0.694 |
| MB both_random | **0.485** | **0.654** | **0.153** | **0.488** | **0.757** |
| MB ta_ta | 0.454 | 0.622 | 0.143 | 0.457 | 0.729 |
| SPECTER2 | 0.273 | 0.417 | 0.083 | 0.277 | 0.533 |

Table 9: Body→TA retrieval in windowed mode. MB both_random gives the best result for every retrieval metric shown here; bold marks the best value in each column.

## E.2 Body→Body Retrieval

| Model | NDCG@10 | MRR | R@1 | R@10 | R@100 |
|---|---|---|---|---|---|
| SB both_random | 0.391 | 0.560 | 0.123 | 0.395 | 0.670 |
| SB anchor_r._pos_ta | 0.366 | 0.532 | 0.115 | 0.369 | 0.641 |
| SB no_ta_ta | 0.389 | 0.559 | 0.123 | 0.392 | 0.670 |
| SB ta_ta | 0.304 | 0.468 | 0.097 | 0.305 | 0.556 |
| MB both_random | **0.457** | **0.631** | **0.147** | **0.458** | **0.727** |
| MB ta_ta | 0.358 | 0.535 | 0.117 | 0.355 | 0.612 |
| SPECTER2 | 0.192 | 0.326 | 0.059 | 0.193 | 0.410 |

Table 10: Body→Body retrieval in windowed mode. This is the hardest scenario, requiring the model to match a body section query against body section candidates. MB both_random leads on every metric; SPECTER2 scores 26.5 points below it on NDCG@10.

Body→Body retrieval is harder than Body→TA for all models because neither query nor corpus contains a title or abstract, leaving the model to rely entirely on partial section content. The **anchor_random_pos_ta** strategy drops notably (0.366 vs. 0.391 for **both_random**) because its citation pairs always use a TA positive, so body–body alignment is learned only through the 20% self-alignment augmentation.

# F    Representation Quality Metric Definitions

Following (Wang & Isola, 2020), we evaluate the embedding space geometry with two complementary properties. **Pair alignment** measures how close citation-linked pairs are: we compute the mean cosine distance between the embeddings of each positive pair's TA views. **Uniformity** measures how well the embeddings utilise the available space:

$$\mathcal{L}_{\text{uniform}} = \log \frac{1}{N(N-1)} \sum_{i \neq j} \exp\big(-t \cdot d(e_i, e_j)^2\big), \quad t = 2 \tag{1}$$

where $e_i$, $e_j$ are L2-normalised embeddings, $d(\cdot, \cdot)$ is the cosine distance, and $N$ is the number of sampled embeddings ($N = 5{,}000$, randomly drawn from the TA embedding pool). The exponential maps each pairwise squared distance to $[0, 1]$: pairs that are far apart contribute $\approx 0$, while pairs that are close contribute $\approx 1$. When embeddings are well-spread, most pairwise distances are large, the mean is close to zero, and the logarithm yields a large negative value. When embeddings are concentrated in a small region, pairwise distances shrink, the mean approaches 1, and $\mathcal{L}_{\text{uniform}} \to 0$.

Alignment is only meaningful when uniformity is sufficiently negative: a concentrated embedding space achieves low alignment trivially (all embeddings are close, so positive pairs are close too), but this does not reflect genuine discriminative structure.

We additionally report **intra-article alignment**, the mean cosine distance between the TA and body embeddings of the same paper, which captures cross-view consistency. As with alignment, this metric is interpretable only when the embedding space is well-spread.

# G TA→TA Full Metrics

| Model | NDCG@10 | MRR | R@1 | R@10 | R@100 |
|---|---|---|---|---|---|
| SB both_random | 0.518 | 0.690 | 0.165 | 0.519 | 0.797 |
| SB anchor_r._pos_ta | 0.528 | 0.697 | 0.167 | 0.531 | 0.807 |
| SB no_ta_ta | 0.517 | 0.689 | 0.164 | 0.519 | 0.797 |
| SB ta_ta | 0.537 | 0.706 | 0.170 | 0.539 | 0.809 |
| MB both_random | 0.535 | 0.703 | 0.168 | 0.537 | 0.801 |
| MB ta_ta | **0.544** | **0.712** | **0.172** | **0.546** | **0.811** |
| SPECTER2 | 0.419 | 0.599 | 0.135 | 0.415 | 0.693 |

Table 11: TA→TA retrieval (body-mode-independent since only title-abstract views are used). Adding body-text supervision does not degrade the standard abstract retrieval setting.