# OpenReview forum: "Beyond Abstracts: Learning Scientific Paper Embeddings from Full-Text Windows"
_ls2n.fr/CORIA-TALN/2026/Workshop/ARTS — ls2n CORIATALN 2026 Workshop ARTS Submission_

### Official Review · Reviewer_Laxc · 2026-04-27

**Mode De Presentation:** Oral

**Confience:**

Oui

**Decision:**

Accepté

**Relecture:**

Points forts :

- Le papier propose plusieurs apports importants pour la modélisation d'embeddings d'articles scientifiques :
	- Un très grand corpus d’articles en texte intégral avec citations, accompagné d’un pipeline et d’un benchmark pour l’évaluation.
	- Une méthode d’entraînement par fenêtres aléatoires combinant apprentissage contrastif et auto-alignement.
	- Un espace de représentations plus discriminant, permettant une meilleure recherche d’articles, notamment à partir du contenu du corps des articles.

- Le papier positionne bien la problématique et les manques actuels.

- Une architecture récente est utilisée (ModernBERT). La baseline me parait également pertinente (Specter2)

- Un souci de réplicabilité des résultats se ressent à la lecture du papier, avec de nombreux détails sur l'implémentation.

- Le corpus et le modèle seront fournis avec la version finale du papier.

Points à améliorer :

- Le résumé pourrait gagner en clarté en définissant plus clairement cette phrase "Sur un benchmark de recherche d’articles fondé sur les citations, cette méthode préserve les performances en TA→TA (0,535 vs. 0,544 NDCG@10) tout en améliorant les tâches à
partir du corps." En effet, soit les informations sont trop précises, et donc nécessitent un peu de contexte (fondé sur les citations ? TA ? TA→TA ? Quelles tâches du corpus du texte ?) soit être plus générique sur les résultats (et donc enlever des détails, en allant directement à la conclusion).

- Le code source aurait pu être fourni complétant les nombreux détails techniques.

- Pourquoi Specter2 n'a pas été adapté avec le corpus collecté ?

Autres remarques mineures :

- Je n'ai pas bien compris pourquoi ModernBERT n'était entrainé qu'avec deux stratégies sur les 4 ? Et la justification du nombre d'époques plus réduit avec ModernBERT ?

**Resume:**

Les auteurs proposent le corpus S2Full composé de 2,5 millions d'articles scientifiques (texte intégral) avec leurs liens de citation. L'objectif de cette collecte est d'améliorer les embeddings d'articles scientifiques. Contrairement aux approches classiques fondées uniquement sur le titre et le résumé, les auteurs entraînent les modèles sur des extraits aléatoires du corps du texte avec un objectif d'auto-alignement. Les auteurs montrent que la méthode proposée conserve des performances comparables pour la recherche s'appuyant sur titres + résumés, tout en améliorant les résultats lorsque les requêtes s'appuient sur le contenu complet des articles. Les meilleurs résultats sont obtenus avec ModernBERT, et des tendances similaires sont observées avec SciBERT. Globalement, l'entraînement sur le texte intégral permet une recherche d'articles plus efficace à partir de formulations variées, au-delà du seul résumé.

---

### Official Review · Reviewer_6PR3 · 2026-04-30

**Mode De Presentation:** Oral

**Confience:**

Oui

**Decision:**

Accepté

**Relecture:**

Le papier est pertinent pour les thématiques de l’atelier ARTS. Ses principaux points forts résident dans la stratégie d’entraînement proposée, ainsi que dans l’ensemble conséquent d’expériences réalisées. La mise à disposition annoncée des deux modèles entraînés, fondés sur SciBERT et ModernBERT, constitue également un apport intéressant pour la communauté.

Mes réserves portent principalement sur la partie expérimentale. Les résultats principaux, présentés dans la Table 4, me semblent relativement attendus dans la mesure où les tâches évaluées, notamment TA->TA et Body->TA, sont très proches des configurations utilisées lors de l’entraînement, en particulier both_random et ta_ta. Les bonnes performances observées sont donc encourageantes, mais elles gagneraient à être discutées au regard de cette proximité entre entraînement et évaluation.

Par ailleurs, la comparaison avec SPECTER2 est informative en tant que point de référence, mais elle me semble peu « fair », car les données d’entraînement diffèrent. Le papier gagnerait à expliciter davantage cette limite, afin de mieux situer la contribution expérimentale et d’éviter une interprétation trop directe des gains observés.

En conclusion, l’article me semble pertinent pour la communauté ARTS, et je serais favorable à son acceptation à l’atelier.

**Resume:**

Cet article présente S2Full, un corpus de 2,5 millions d’articles scientifiques en texte intégral, enrichi de liens de citation, ainsi qu’une stratégie de masquage aléatoire pour l’entraînement de modèles de type BERT. Les expériences menées sur une tâche de recherche de citations montrent que cette stratégie d’entraînement, appliquée aux modèles backbone SciBERT et ModernBERT, est efficace et permet d’obtenir de meilleures performances que SPECTER2 avec l’adaptateur PRX.

---

### Decision · Program_Chairs · 2026-05-07

Accept (Oral + Poster)